# The Digital Bytes Project: Digital Storytelling as a Tool for Challenging Stigma and Making Connections in a Forensic Mental Health Setting

**DOI:** 10.3390/ijerph20136268

**Published:** 2023-06-30

**Authors:** Caroline Lambert, Ronnie Egan, Shelley Turner, Miles Milton, Madeleine Khalu, Rishona Lobo, Julia Douglas

**Affiliations:** 1Social Work and Human Services, School of Global, Urban and Social Studies, Melbourne City Campus, RMIT University, Melbourne, VIC 3000, Australia; 2Forensicare (The Victorian Institute of Forensic Mental Health), Fairfield, VIC 3078, Australia

**Keywords:** forensic mental health, digital storytelling, social work

## Abstract

This article reports on the findings of a study that explores the utility of digital storytelling as a narrative practice and learning tool for social work in an Australian secure forensic mental health hospital. The short digital stories, or Digital Bytes Project, centered on capturing the lived experience, hopes and perspectives of the hospital’s service users by giving voice to their experiences through digital technology. The project was collaboratively designed and co-delivered with social work students, hospital staff, and service users. It aimed to not only destigmatize people with lived experiences of mental distress and criminal justice system involvement but also to give staff and students further insights into understanding who they are working with. Through a series of 11 semi-structured, one on one interviews, this research aims to explore social work student and forensic mental health staff experiences and perceptions in relation to the utility and impact of these digital bytes, reflecting on how the prototype bytes may have impacted their learnings, or practice, including how they then interact with service users. This research investigates how these digital bytes could be used further within forensic mental health organisations and contexts. The research findings demonstrate the overall value of digital bytes in challenging different kinds of stigma, shifting power dynamics and staff perspectives; strengthening rapport and understanding through enhancing engagement and sharing power between students, staff, and consumers; as well as providing insight into the utility of digital bytes for learning and making connections between theory and practice. The preliminary findings from this research suggest the need for greater accessibility, integration, and consideration of these digital tools, with their potential to be translated across multiple human service sectors.

## 1. Introduction 

Digital storytelling (DST) is the practice of people using digital tools to produce and share their stories. These stories are often powerful, use emotionally engaging images, and cover various digital formats. Digital stories may include photographs, animation, video, sound, music, and the storyteller’s voice to convey an intimate insight into a lived personal experience [1,2]. Chan and Sage [3] suggest that as a narrative practice, DST has the potential for social workers to function as a change-making strategy at both the micro and macro levels of practice. It may facilitate positive client outcomes through crafting, reflecting on, and possibly sharing a preferred self-narrative, and it may act as a counter-narrative, challenging broader, unhelpful social constructions. While the benefits and risks of using DST to support mental health recovery have been relatively well explored [4,5,6] its utility within a forensic mental health context appears unexamined. This is significant, given the so-called ‘dual stigma’ experienced by people involved in forensic and mental health systems [7,8]. This article explores the potential utility of digital storytelling as a learning tool and narrative practice with forensic mental health consumers through an analysis of a student-led social work project, Digital Bytes. The terms ‘consumers’, ‘clients’, ‘patients’, and ‘service users’ are used interchangeably throughout this article to reflect the contested nature of language and labels and the various terminology preferences of the people involved in the project.

## 2. The Digital Bytes Project

The authors’ interest in digital storytelling and forensic mental health developed as the result of a student ‘studio’ in 2017, which was run collaboratively by the social work departments at RMIT University and Thomas Embling Hospital. The project was driven and overseen by the lead author while jointly employed as Forensicare Industry Fellow at RMIT University and as social work and lived experience educator at Forensicare, with support from the Chief Social Worker and Lead Social Worker at Thomas Embling Hospital. The student studio was based off-site, creating a new space that was differentiated from their work or study environments. The students enrolled in the studio process followed established design thinking stages, such as ideation, exploration, consultation, and prototyping, to imagine and develop a new solution to the well-established problem of stigma [9]. Thomas Embling Hospital is a secure psychiatric hospital in Melbourne, providing acute and continuing care programs for over 130 people [10,11]. People are generally admitted to the hospital from the criminal justice system under the Crimes [12], Mental Health Act 2014 or the Victorian Sentencing Act 1991, with their length of stay averaging around seven years [10,11]. The hospital is operated by Forensicare (The Victorian Institute of Forensic Mental Health), the state-wide specialist provider of forensic mental health services in Victoria, Australia [13].

Since developing the student ‘studio’, social work students completing work-integrated learning placements at Thomas Embling Hospital have contributed to the Digital Bytes Project by co-creating stories with consumers using digital methods. The aims of the Digital Bytes Project can be summarised as follows:To provide a learning opportunity for social work students on placement at Thomas Embling Hospital to develop engagement skills.To provide consumers with a chance to direct their narrative, which could guide staff on engaging with them; andTo reduce stigmas that forensic mental health service users might be experiencing.

Using a series of design processes [14], the social work student studio lays the foundations for the development of consumer digital byte prototypes to store them within the consumer electronic records system at the hospital as a way for clinicians to access and engage with the bytes regularly. The student studio design process, and the participatory methods employed in the byte production, spoke to the intentional focus on congruence with the social justice aims of social work in forensic mental health or to a ‘critical clinical’ mental health social work practice [15].

The ‘bytes’ are three to five minutes long digital stories created by the consumer with student support. The focus of the digital story is determined by the consumer, with the bytes produced focusing mainly on a consumer’s hopes for the future, recovery, and self-identity. Several digital bytes contained consumer narratives around what they liked, and disliked, what gave them purpose and what they would dream about in a life post-release. Some consumers shared their artistic achievements, songs, poetry, or photography. All the consumers shared ways that staff could build rapport and connect. This link between consumer/survivors, ‘madness’ and artistic methods of communication is well documented throughout history [16,17]. Consistent with a narrative approach, the bytes are intended to be strengths-based and recovery focused, and to facilitate the consumer’s expressed sense of positive identity. The bytes spotlight clients’ strengths and can provide small moments of service user autonomy and choice in an otherwise relatively restrictive statutory environment. The underpinning theoretical and conceptual frameworks can thus be described as narrative, anti-oppressive and strengths-based. These align with recovery principles and establish social work practice, and may help understand, challenge and shift established hierarchies.

### Forensic Mental Health, Stigma, and Social Work

Forensic mental health services operate at Australia’s nexus of mental health and criminal justice systems. People who use these services, and their families and carers, typically experience stereotyping, prejudice, and discrimination. Their stigmas include public or social stigma and internalised and structural stigma [18,19]. This study uses Groot’s definition of public or social stigma as a multidimensional psychological construct that “…encompasses negative implicit and explicit cognitive, affective and behavioural responses to persons living with, or more abstractly, the concept of, mental ill-health” [20] (p. 7). According to Groot, “This stigma involves a deep discrediting and devaluing of persons living with mental ill-health by others” [20] (p. 7). Users of mental health services might also suffer from internalised stigma, defined as negative attitudes and stereotypes that people living with mental illness hold about themselves [21]. Structural stigma exists at the broader, macro-social level [22] and emphasises the role of institutions and cultural ideologies in perpetuating bias and discrimination [19].

For people involved in forensic mental health systems, these multiple stigmas are further compounded [23] and often fostered by prejudicial and sensationalised media coverage that stereotypes consumers as consistently dangerous, violent, and unpredictable people [7,8,24,25]. O’Donahoo and Simmonds describe the broader social context for forensic mental health service users as an environment hallmarked by stigma, discrimination, and oppression, with ‘zero tolerance’ for offending [7]. The negative impacts of stigma and discrimination are also reflected in structural and systemic responses, which often exclude, and further marginalise people living at the intersection of mental distress and offending behaviour.

Forensic mental health social workers balance so-called ‘risk’ and ‘recovery’ tensions in their work and consider the various and sometimes competing needs of consumers, their families and carers, and the general community [13]. They assess and respond to service users’ risk of harm to themselves and others while championing their strengths and human rights. Forensic care social workers adopt “a collaborative, strengths-based approach to working with consumers, families and carers… [and]… aim to address both individual needs and risks, while also challenging structural inequalities that hinder personal recovery and contribute to the stigmatisation of people in forensic mental health” [13]. Addressing stigma through strength-based approaches aligns with the *AASW* (Australian Association of Social Workers) Practice Standards for Mental Health Social Workers [26] (pp. 8–9), specifically: Standard 1.2 (b.), ‘Challenges stigma and discrimination’, and Standard 1.3 (d) […] challenges stigmatising attitudes and discrimination. It also aligns with Forensicare’s model of recovery-focused care (Forensicare, 2021–2026) [27], which promotes “hope, social inclusion, personalised care and self-management” and aims to “reduce stigma and improve community understanding of people living with a mental illness and offending behaviours”.

## 3. Methods

The current study explores social work students’ and practitioners’ perceptions of the utility of Digital Bytes as a learning tool and narrative practice with forensic mental health consumers. It involves students and social workers interacting or engaging with the Digital Bytes Project at Thomas Embling Hospital. This study received approval from the RMIT HREC, approval number 23390S. Due to the small number of participants and the identification of institutions, we chose not to include demographic data that could increase the possibility of identifying participants.

A scoping review was conducted of peer-reviewed literature published between 2011–2021 to examine how digital storytelling (DST) is being used to address experiences of stigma and discrimination for mental health service users, particularly forensic mental health service users [28]. The timeframe was chosen to capture current material, noting the pace of evolving technologies. The scoping review included 11 peer-reviewed articles, with findings indicating an overall positive impact and utility of digital storytelling for mental health consumers in reducing self-stigma and ameliorating stigmatising attitudes of both staff and the broader community [28]. However, no research literature was found that focuses on the impact of digital storytelling on a forensic mental health population. The findings of the scoping review, and a peer-reviewed text that explores the use of DST for training and educational purposes, helped inform the development of interview questions for the current study.

Semi-structured, one-on-one interviews were conducted with eleven participants about their experiences and perceptions of the utility of Digital Bytes as a learning tool and narrative practice. All participants were involved with the Digital Bytes Project at various times over the project’s six-year life span. Recruitment to the study was via internal emails to key discipline leads through a purposive recruitment strategy. Although staff from multiple disciplines (i.e., nursing, occupational therapy, psychology) were invited to participate, only social workers and one designated lived experience worker agreed. Of these eleven participants, one was an RMIT social work student, nine were a Forensicare social worker, and one was a Forensicare lived experience worker. Seven participants were current Forensicare staff members; one was a former staff member; two identified as ‘students’ at the time of involvement in the digital bytes and had since transitioned into employment at Forensicare; and one participant was a social work student on placement at Thomas Embling Hospital from RMIT University. All participants had varying degrees of engagement with the Digital Bytes Project. For the students involved in the project, some contributed to research to support the project, and others were engaged through co-creating digital bytes directly with consumers. The interviews were recorded, transcribed and de-identified before the commencement of the analysis. Braun and Clarke’s six-step framework for conducting a thematic analysis was used to analyse the interview transcripts [29]. Our researchers analysed and coded each transcript independently, then compared codes once all interview transcripts had been coded to identify key themes and subthemes. Coders reached a consensus on the codes identified in the transcripts, increasing rigour and reliability.

## 4. Findings

Three interconnected vital themes were identified that addressed the question of the utility of Digital Bytes as a learning tool and narrative practice:Challenging stigmaStrengthening rapport and understandingMaking connections

Some subthemes were also identified as part of these overarching themes, sometimes interconnected with others.

### 4.1. Theme 1—Challenging Stigma

The critical theme identified from the data was that Digital Bytes challenged the stereotypes and prejudices about forensic mental health consumers, thereby challenging associated stigmas. The Digital Bytes Project allowed social workers and students to engage meaningfully with consumers in a secure hospital through digital storytelling. In turn, this appeared to have an overall ‘humanising effect’ that served to recharacterize staff and students’ understandings and perceptions of consumers. More specifically, the interviewee responses indicated that Digital Bytes challenged both individual and structural forms of stigma.

“*[Digital Bytes] can give you a sense of [consumers] as a person… a more holistic view of them, which is automatically going to reduce stigma and discrimination from staff*”(Participant 2).

“*I didn’t appreciate… how meaningful it would be for the patient to have the opportunity to tell their story to staff… When you work within a medical model, much of what you do is mired in either a risk or just the core tenants of your role. Watching the [Digital Bytes] videos… humanises…patients much more than what is already in our practice. So, I think it’s a really important addition to what we do.*”(Participant 11).

“*I think there’s a lot of value underlying [the digital bytes] project… I think it humanises a cohort of people that we work with, that we tend to see more in a forensic lens…*”(Participant 6).

Interviewees explained that the Digital Bytes Project challenged stigma in two key ways:Co-creating a ‘byte’ increased meaningful engagement between staff or students with consumers, which in turn,Helped challenge and shift any negative preconceptions that staff and students held about forensic mental health consumers.

For example, some participants spoke about the Digital Bytes Project providing a more holistic perspective of the consumer than just their main (index) offence or presenting health issues.

“*I think for a lot of staff in the hospital… they need to talk about medication, or they need to know about the index offence, or they think they need to know about the index offence. So there’s a real focus on that… I think it’s important to get to know people on a human level…to know a patient outside of their status as ‘patient’…and perhaps it’s Digital Bytes that allows that*”(Participant 2).

“*The value of digital storytelling is that you are able to see the person beyond their index [offence]. Seeing them just as a person, just as any other human being, and that they have dreams, goals and hopes just like anyone else. I think [Digital Bytes] has the potential to reduce stigma and… get the treating team to know the patient… on a more personal level*”(Participant 9).

“*…so I think that it’s an opportunity for maybe someone with a bad rap … to be able to tell … their side of the story in terms of who they are as a person and help to try and change perspectives…*”(Participant 6).

“*I think it gives you a more rounded perspective of the individual. I think it gives you some insight into their inner world and … life beyond being a patient… they have other interests, abilities and wants and needs which are not really captured…fully*”(Participant 10).

“*There was one [Digital Byte] that I watched…that was quite worthwhile because…this patient had a particularly… for want of a better word ‘notorious’ history and background… I had no idea that this person enjoyed… cooking and sculpture… I feel like that sort of knowledge. I wouldn’t have known… unless I watched [their Digital Byte].*”(Participant 11).

The Digital Bytes offered more personalised information and details about consumers that were not captured within case notes or professional documentation, and which allowed staff and students a better, fuller understanding of their lives. Interviewed social workers highlighted how official patient or ‘criminal’ records often have a reductionist and dichotomous effect, while digital storytelling allowed an opportunity to build a fuller picture of the consumers’ skills and possibilities:

“*If you pick up a lot of file information, it’s very categorical info[rmation]. And… can be quite reductionist at times and so I think the digital Bytes do have a nice effect of seeing the story itself, you see someone talking about themselves, and it’s received in a different way. So, we could learn about how someone might interact and how they see themselves. Some of their skills and presentation can come out and what their inner life is like, rather than relying on file information*”(Participant 4).

“*I think [Digital Bytes] is…another way to show a different side to the patient… Instead of reading documents created by someone else on the patient, they’re able to be like, “This is me. This is what I want you to know about me”, and I think that’s really valuable in itself*”(Participant 3).

“*[A]s clinicians we generally look at our clients on paper…and the negative side of them can be portrayed because we are such a risk-averse organisation. Whereas the Digital Bytes… portrays the client from their perspective of putting positive things across*”(Participant 8).

Watching and participating in the creation of Digital Bytes shifted some staff and student perspectives from holding somewhat negative perceptions about consumers to viewing them as individuals with dreams, hopes and goals rather than just someone who offended them. Through the co-creation of these Digital Bytes, service users’ interests and abilities, preferences and hobbies were discovered, increasing relatability and challenging stigma and discrimination.

### 4.2. Theme 2—Strengthening Rapport and Understanding

Developing a positive relationship or therapeutic alliance is fundamental to forensic mental health staff and students working well with consumers [30]. Participants spoke about the nature and importance of such relationships in the secure hospital setting, noting that the Digital Bytes Project was useful for building and strengthening rapport and understanding with consumers. This occurred through:The processes of co-creating the ‘bytes’.staff and students’ exposure to Digital Bytes, andhow staff and students approached their engagement with consumers upon viewing their digital story or ‘byte’.

In other words, the participants found co-creating, watching, and applying the learnings from a ‘byte’ useful as a narrative, recovery-focused practice that promotes engagement and power-sharing between staff or students and consumers [31,32]. Indeed, engagement and sharing power were identified as two subthemes of strengthening rapport and understanding.

Co-creating a digital story or ‘byte’ with consumers gave staff and students the time and space to build trust and strengthen rapport with consumers and gain a deeper understanding of their experiences and perspectives. In turn, these insights informed and tailored the recovery-focused care provided by staff and students.

“*[The Digital Bytes] project [creates] an opportunity for new staff to really…get to know someone and get some conversation starters …to build rapport. It’s…also a chance…for patients to…play an active role in [their] engagement with the treatment team and…the recovery process*”(Participant 6).

“*I see it [Digital Bytes] as very valuable to staff members to get to know the patient*”(Participant 1).

“*[I]t just builds that really good rapport, and that bond between the staff and the consumers trust you*”(Participant 5).

“*[Before watching or co-creating a Digital Byte] I didn’t know…the patient himself. He was able to talk about… [h]is goals and his hopes of leaving the hospital and what he hopes to do, what he wanted to be… [which led me to] having a better understanding of who he was as a person*”(Participant 9).

“*I think one of the big parts of just mental health practice generally, but for all disciplines, is the relational security and rapport that you have with your clients. Uhm, knowing their likes and dislikes and what’s going on for them in their internal world, and that doesn’t even necessarily mean their delusions or auditory hallucinations, but it could be when they’re feeling sad or depressed, or what they like to or don’t like. I feel knowing these things makes your practice better and leads to better outcomes for patients… I feel like the Digital Bytes program, based on what I’ve seen, is useful in that regard because it bridges a divide that would otherwise maybe take a little bit longer if you were to just sort of inquire about these sorts of things. Basically, it’s like being in a room without the lights on. You’re searching around looking, trying to build a rapport with the patient whereas a program, [Digital Bytes]…gives you a head start*(Participant 11).

The Digital Bytes Project appeared to positively impact forensic mental health staff and students’ approach and ability to engage with consumers, as it offered a creative, consumer-driven method of engagement that opened up new understandings about the consumer and potentially new opportunities for recovery-focused engagement.

“*I see [Digital Bytes] as very valuable for staff members to get to know the patient… and give them talking points so they’re able to connect instantly instead of kind of awkward small talk that is not that valuable to the consumer, for the consumer to have an outlet…*”(Participant 1).

Watching a ‘byte’ and its informative nature was seen as an avenue for developing an understanding of consumer needs, an opportunity to build rapport more rapidly with consumers and for forensic mental health social workers and students to tailor their practice approaches accordingly. Engaging with a short yet informative video was suggested to be more time efficient and human-centred than reading through dry documentation.

“*[T]here were…nurses who had worked on the unit for ten years, saying: oh, I never knew that about that person until they have seen [their] Digital Byte. […] Video is…a really powerful tool… It allows people to engage with storytelling in a simpler way. [For] time poor [staff]… having…a two-minute video to watch is much easier than reading…a two-page document*”(Participant 2).

“*[The] patients have other interests and other desires and wants beyond which was written down on case file notes…*”(Participant 10).

“*Watching the [Digital Byte] video… was really worthwhile in terms of learning what type of things I’d be able to engage with them on if I ever had to. It… also broadened the scope of actual recovery-based activities that could be done together*”(Participant 11).

Through engaging with Digital Bytes, forensic mental health staff and students could develop a new understanding of consumers, even when some clinicians had worked with certain participants for extended periods. Participants spoke to staff, developing new understandings that encapsulated consumer hopes and goals for the future. This suggests that Digital Bytes can enhance therapeutic relationships, direct therapeutic focus, and potentially lead to better consumer outcomes aligned with their personal goals.

The Digital Bytes Project was seen by participants as a way of cultivating consumer agency and redressing, to some extent, the inherent, uneven power structures that exist within forensic or criminal justice contexts, as well as mental health systems [23]. In line with emancipatory and participatory practices, such as co-production and co-design, the Digital Bytes Project is built upon the notion of shared power. Participants described power sharing as promoting client-driven *engagement* between consumers, staff, or students.

“*[The]content of digital bytes is driven by the consumer. So, it’s really…a true collaboration in its truest sense*”(Participant 10).

“*I think [Digital Bytes] is very valuable because…[i]t’s something [consumers] have control over… a form of self-expression.*”(Participant 3).

“*They’re [consumers] telling us how to engage with them…Tell[ing] their staff member about themselves in their own words*”(Participant 7).

“*Digital Bytes…can be a very powerful tool. [It gives] consumers the voice to express themselves, to show their personality…*”(Participant 9).

“*[Y]ou can speak with a client in a casual…setting rather than a structured formalised assessment or an interview room, I think breaks down power structures…*”(Participant 11).

Through Digital Bytes, consumers played an active, central role in the digital storytelling design and narrative, shared with forensic mental health social workers, lived experienced staff, and social work and lived experience students. This degree of power-sharing, facilitated by the Digital Bytes Project, was seen as critical in the forensic mental health setting. Like other statutory social work settings with involuntary or mandated clients, there are limited opportunities for exercising consumer choice and control. 

### 4.3. Making Connections

Participants defined the Digital Bytes as examples of learning tools which allow social work staff and students to make direct connections between theory and practice and to acquire and develop important interpersonal and communication skills needed to co-create the ‘bytes’ with consumers.

Digital storytelling allows staff to understand the consumer’s perspective and take strengths from their narrative [33]. Many participants explicitly identified narrative and strength-based approaches as underpinning Digital Bytes and aligning with recovery principles, with three participants referencing the use of Narrative Theory, while seven participants stated that the Digital Bytes intensively used a strengths-based approach.

“*I think that…digital bytes has like a strengths-based theory or perspective behind them which then really measures with the recovery model as well*”(Participant 8).

“*[Digital bytes incorporates a]”…strengths-based approach [by] really looking at the person’s capacity and their aspirations and seeing how you can… incorporate that into their recovery process*”(Participant 9).

“*[Digital Bytes involves]…using a narrative approach as well in terms of getting someone to tell their story…then using that as an opportunity to understand their experiences*”(Participant 6).

“*It’s about recovery, it’s about narrative. It’s about their narrative, not ours*”(Participant 7).

The interviewees stated that the Digital Bytes Project created a space that allowed staff to look at the person in his/her/their capacity and view consumer aspirations incorporated into their recovery process. Participants mentioned that the Digital Bytes Project is recovery-oriented in a way that includes the consumers in their path to recovery, and strengths-based approaches facilitated staff and students learning about the consumer’s recovery goals, hopes, aims and aspirations.

While many participants stated that the Digital Bytes were impactful, they were also flagged as not easily accessible, and this limited the potential utility and impact of the digital bytes for influencing everyday practice in the forensic mental health context. Indeed, accessibility was identified as a subtheme of the overall theme of making connections, as the lack of accessibility of the Digital Bytes—or the lack of ease with which people could locate the ‘bytes’ within the technological practice setting—hampered connection-making. Two participants mentioned that the unit staff did not know where the ‘bytes’ existed, while nine noted that many staff do not know they exist.

Some participants mainly focused on the idea of making the Digital Bytes accessible to new staff and employees, suggesting they could all be added to the patient information management system (PMI) in the hospital for staff to use while providing clinical handovers to the next treatment team.

“*Once it was filmed and put on PMI [Patient Management Index—the electronic system used to store patient information], there was some really positive feedback from the consultant and some of the nursing staff on the unit about it…There was some curiosity and some interest, but people just didn’t have access…[It’s] not as effective as people hoped it would be.*”(Participant 2).

One participant highlighted the potential for further building on this project with the right kind of attention given while designing the output of the project.

“*[At present the impact of digital bytes is] “minimal but has the potential to be quite significant…*”(Participant 1).

Indeed, all the participants noted the project’s potential, citing accessibility as the biggest barrier to reaching this, thus lessening their impact on staff and consumers.

## 5. Limitations and Future Directions

A key limitation of this research is that the sample of participants did not reflect the multi-disciplinary workforce in forensic mental health, as participants were affiliated with either the social work profession or the lived experience workforce. However, this study provides useful insights into the utility of the Digital Bytes Project for social work practice in forensic mental health. It highlights the natural alignment between the social justice values underpinning social work and lived experience and the narrative foundations of digital storytelling. This research also does not explore the utility of the Digital Bytes Project for consumers, their families, or carers, which is essential for measuring or understanding utility more meaningfully and holistically.

## 6. Discussion

The project was co-created between social work students, staff, and service users, aiming to contribute to the de-stigmatisation of people with lived experiences of mental distress and justice system interaction and to give students and staff further insights into who they are working with. It is, perhaps, an indictment on the dominant medical model that current mental health practices, paradigms and models of care may not be equipped to facilitate a richer discovery of personal consumer perspectives [34,35]. This study explores both the utility and potential of the Digital Bytes Project as a narrative practice and learning tool for social work in a forensic mental health setting. It also aims to investigate how digital bytes could be used further within a forensic mental health context. The research findings demonstrate the overall value and utility of digital bytes in challenging stigma, shifting staff perspectives, and strengthening rapport and understanding by promoting engagement and power-sharing between students, staff, and consumers. The findings further demonstrate the utility and potential of Digital Bytes as a learning tool to make connections between theory and practice and shape everyday practice in a forensic mental health setting. This discussion will integrate the findings with the available literature on digital methods and then identify the implications of the current research on key stakeholders within the forensic mental health setting.

From the findings in this research, the digital bytes can potentially reduce the stigma of consumers in the forensic mental health setting at two levels: by shifting staff attitudes and changing internalised consumer attitudes through participation in digital storytelling. This reduction has occurred primarily through the positive humanising of consumers through DST, seeing consumers beyond their ’index offence’ as relatable people with hopes and dreams. Such shifts in staff attitudes are in line with the literature, which identifies the role that digital storytelling can play in challenging the misperceptions and stereotypes that contribute to stigma and discrimination of people living with mental distress or illness [6,36]. It also aligns with Chan and Sage’s conceptualisation of DST as a narrative practice, which has the potential as a change-making strategy for social work at both the micro and macro levels of practice [3].

Participants spoke about viewing consumers on paper through a “forensic lens” within a risk-averse organisation. De Vecchi et al. highlight that the participatory process and digital method push back on the traditional medical model hierarchy, recognising consumers as experts and disrupting the ascendant narrative of control by others [37]. In this way, the Digital Bytes Project could function as a broader counter-narrative that emphasises clients’ strengths and potential rather than risks and deficits [38]. Further research exploring DST as a tool to counter the dominant medical or disease models is recommended. Other research highlighting the impact of DST on community members also notes the humanising and personalising responses [39,40], which were congruent with participants’ narratives in the current research.

The student participants in this research who co-created the digital bytes noted the meaningfulness for the consumer of having the opportunity to tell their story. The reduction in self-stigma, and an increase in positive self-identity using participatory digital narratives, have also been noted in the literature which details consumer accounts. This resonates with Sapouna and Pamer’s research [41], which identified that many mental health consumers involved in digital storytelling projects spoke of broadening their self-concept beyond a narrowly defined ‘patient’, or someone with ‘ill health’. Such accounts provide a positive counter-narrative disrupting the dominant stigmatising narratives often present in stigmatising popular media representations [42,43,44], particularly for consumers, families and carers involved in forensic mental health.

All participants spoke about the Digital Bytes Project enhancing opportunities to strengthen rapport and understanding between practitioners, students, and consumers. This occurred through consumer-driven engagement and a sharing of power in making and sharing the digital bytes and in the interactions with consumers about their digital stories.

Engagement between practitioners and consumers was described as “consumer-driven” and allowed practitioners to connect with a consumer-led narrative rather than their perspective being filtered through formal documentation and the lens of the expert/system. The research indicated that their stories were a more holistic representation of them rather than a narrow focus on offending behaviour and acute mental illness. The student participants in creating the digital bytes noted that the process allowed consumers to choose how to depict and define aspects of themselves and to externalise their thinking into the digital space. This mirrors Chan and Sage’s reporting of the value of digital media for this approach [3].

The final way participants spoke about strengthening rapport and understanding was through shifting power dynamics between consumer, student, and practitioner. With limited choice within the forensic setting, participants noted that digital bytes gave the consumer power to narrate their story. From the consumer perspective, De Vecchi et al., report that mental health consumers felt power in capturing their journey and controlling the narrative of their own story [37]. Link and Phelan (2001) argue that inequitable power distribution is a necessary condition of stigma existing and thriving: any process that heads toward a rebalancing of uneven power is a vital factor in potentially reducing stigma and discrimination [22]. It is the potential utility of digital bytes in providing the opportunity to balance—even temporarily or incrementally—the power dynamics between staff and consumers that highlights their value. In this way, Digital Bytes, as a participatory method, promote the core principles of co-design that are arguably shaping contemporary mental health and forensic systems [22,45].

The findings about the utility of digital bytes as a way of making connections present both positive and negative experiences for participants. The positive experiences reflect the value of the bytes as an essential learning tool that allows students and practitioners to make explicit connections between theory and practice, particularly narrative theory and strength-based perspective as drivers of recovery-focused practice [33,45]. The reflected learnings depended on whether the participant was a staff member or a student. For staff, they valued learning about consumer aspirations/recovery goals and saw them as a valuable way of acquiring skills relating to strengths-based work. Students, particularly those involved in producing the bytes with consumers, spoke about their learning associated with integrating theory with practice, developing shared purpose with consumers and a genuine collaboration in the production process. Johnston et al. also highlighted how digital storytelling could help foster students to reflect and recognise their internal stigma [14].

The reported negative experiences were more systemic and related to difficulty accessing or knowing about the digital bytes stored in the electronic patient information management system. This indicated a lack of overall integration of the Digital Bytes into existing forensic mental health practice. Some of this can be attributed to the disruptions to service delivery and priority changes during the project period due to the COVID-19 pandemic [46]. However, the lack of connection to daily practice also highlights an essential limitation of attaching a ‘once-off’ consumer story to the clinical records system. To retain relevance, consumers need to be able to update or re-narrate their ‘byte continually’. This limitation of adding the byte to the clinical record without a clear review process is consistent with the process required for reviewing and updating assessment reports.

## 7. Conclusions

Finally, the Digital Bytes Project has demonstrated an important degree of utility for social work in forensic mental health, both as a tool for making connections and for narrative practice. This is evident from the overall positive findings about the project, derived from the interviews with social work students, social workers, and a lived experience worker about their experiences of involvement in the Digital Bytes Project. Perspectives of consumers, families and carers, and staff from other disciplines remain untested. Future research should focus on these perspectives, particularly their views on the meaningfulness and genuineness of any power-sharing. Moreover, as a student-led project, Digital Bytes may have reached its capacity to impact practice and challenge stigma at the micro or macro levels. Learnings from this project are recommended to inform training and practice development for forensic mental health staff to ensure better integration of DST as a narrative practice and multi-disciplinary learning tool to support personal consumer recovery. At the macro-level, the learnings from this project could form the basis for a broader, interdisciplinary, and co-designed project that aims to challenge the dominant, stigmatising narratives about forensic mental health consumers and their families and carers.

## Data Availability

Datasets regarding the Digital Byte Project are currently unavailable.

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
