# Peer review of "The Digital Bytes Project: Digital Storytelling as a Tool for Challenging Stigma and Making Connections in a Forensic Mental Health Setting"

_ijerph, 2023, doi:10.3390/ijerph20136268_

Round 1

Reviewer 1 Report

Hi authors,

I have attached my report to this review. Well done. A bit more work on the ms and you should be there.

Reviewer

Reviewer 2 Report

Thank you for sharing this paper about the Digital Bytes project.

As a reader, I’m interested to learn more about the design process the social work student employed to get this project up and running. More detail here would simply be interesting and might offer insight into the backstory behind this work.

Also, as you point out, video as powerful tool. It would be wonderful to have access to one or more videos to accompany the paper. Or, in the absence of the videos, please consider including a section that describes one or more video. The content of the videos seems crucial to understanding their impact.

It would also be useful to apply some critical analysis. For example, I’m left wondering: why does it take a digital storytelling project for staff to learn about the people accessing care in this facility? Also, it would be useful to know the history of this institution. Recently disciplines such as mad studies are including the histories of institutions such as the Thomas Embling Hospital. For further context, consider linking this digital storytelling work to larger histories of madness that include the art and cultural creation of such labelled people (consumers, clients, survivors, etc.).

Reviewer 3 Report

This article reports on findings of a study that explores the utility of digital storytelling as a narrative practice and learning tool for social work in an Australian secure forensic mental health hospital. The short digital stories, or Digital Bytes project, centred on capturing the lived experience, hopes and perspectives of the hospital’s service users by giving voice to their experiences through digital technology. The project was collaboratively designed and co-delivered with social work students, hospital staff, and service users. It aimed to not only destigmatise people with lived experiences of mental distress and criminal justice system involvement, but to also give staff and students further insights into understanding who they are working with.

The authors provided a work that was informative, well written, and provides a solid foundation for future research. I enjoyed reading this manuscript, and recommend the authors thoroughly read the manuscript and correct any spelling and/or grammatical errors. It is in the spirit of improving the manuscript that I offer the following comments and/or questions:

Change your title to:

The Digital Bytes Project: Digital Storytelling as a Tool for Challenging Stigma and Making Connections in a Forensic Mental Health Setting

On Page 1, you wrote: Digital Bytes project 

Change all to: Digital Bytes Project [Sometimes you spell “project” and other times you spell “Project” – Make it consistent throughout]

On Page 1:

Change “centred” to “centered.”

On Page 2, you wrote:

While the benefits and risks of using DST to support mental health recovery have been relatively well explored (see De Vecchi et al. 2016, Botfield et al. 2017, and de Jager et al. 2017), its utility within a forensic mental health context appears unexamined. This is significant, given the so-called ‘dual stigma’ experienced by people involved in both forensic and mental health systems (see Marklund, Wahloos, Loori & Gabrielsson, 2020; West, Yanos & Mulay, 2014; O'Donahoo & Simmonds, 2016).

Change to:

While the benefits and risks of using DST to support mental health recovery have been relatively well explored (see Botfield et al., 2017; de Jager et al., 2017; De Vecchi et al., 2016), its utility within a forensic mental health context appears unexamined. This is significant, given the so-called ‘dual stigma’ experienced by people involved in both forensic and mental health systems (Marklund et al., 2020; O'Donahoo & Simmonds, 2016; West et al., 2014).

·        Per the 7th Edition of APA, in-text citations should be in alphabetical order.

·        Per the 7th Edition of APA, when citing 3 or more authors, you should use the surname of the first author, et al., and then the year.

·        Make sure that all citations that are in the manuscript are also on the Reference page [See Botfield et al., 2017; de Jager et al., 2017; De Vecchi et al., 2016; Marklund et al., 2020; O'Donahoo & Simmonds, 2016; West et al., 2014]. Several citations were in the manuscript but not on the Reference page.

NOTE: You made these errors several times in the manuscript. Please correct these errors.

On Page 3, you wrote:

A scoping review was conducted of peer reviewed literature published between 2011 – 2021, to examine how digital storytelling (DST) is being used to address experiences of stigma and discrimination for mental health service users, particularly forensic mental health service users (Lambert, Egan, Turner & Martin, 2023 - under review).

·        Why did you start in the year 2011? Is this the year that digital storytelling (DST) was first used among mental health service users, specifically forensic mental health service users?

On Page 4, you wrote:

Braun and Clarke's (2006) six-step framework for conducting a thematic analysis was used to analyse the interview transcripts. Our researchers analysed and coded each transcript independently, then compared codes once all interview transcripts had been coded to identify key themes and subthemes.

·        This citation (Braun & Clark, 2006) should be added to the Reference page.

·        Make sure that all citations that are in the manuscript are also on the Reference page.

On Page 4, you wrote:

Our researchers analysed and coded each transcript independently, then compared codes once all interview transcripts had been coded to identify key themes and subthemes.

·        Did you establish reliability between the coders? If so, what was the reliability percentage? If not, didn’t you establish reliability between the coders?

On Page 4, you wrote:

This stigma involves a deep discrediting and devaluing of persons living with mental ill-health by others” (Groot, 2021, p.7).

Change to:

This stigma involves a deep discrediting and devaluing of persons living with mental ill-health by others” (Groot, 2021, p. 7).

·        I added a space between p. and 7.

On Page 8, you wrote:

“There was one [digital byte] that I watched…that was quite worthwhile because…this patient had a particularly… for want of a better word ‘notorious’ history and background… Watching the video… was really worthwhile in terms of learning what type of things i’d be able to engage with them on if I ever had to. It… also broadened the scope of actual recovery-based activities that could be done together. I had no idea that this person enjoyed… cooking and sculpture… I feel like that sort of knowledge, I wouldn’t have known… unless I watched [their digital byte]. Activities like that… where you can speak with a client in a casual…setting, rather than a structured formalised assessment or an interview room – i think breaks down power structures…” (Participant 11).

Change to:

“There was one [digital byte] that I watched…that was quite worthwhile because…this patient had a particularly… for want of a better word ‘notorious’ history and background… Watching the video… was really worthwhile in terms of learning what type of things I’d be able to engage with them on if I ever had to. It… also broadened the scope of actual recovery-based activities that could be done together. I had no idea that this person enjoyed… cooking and sculpture… I feel like that sort of knowledge, I wouldn’t have known… unless I watched [their digital byte]. Activities like that… where you can speak with a client in a casual…setting, rather than a structured formalised assessment or an interview room – I think breaks down power structures…” (Participant 11).

On Page 9, you wrote:

[Digital bytes incorporates a]”...strengths-based approach [by] really looking at the

person own capacity and their own aspirations and seeing how you can… incorporate that into their recovery process” (Participant 9).

Change to:

[Digital bytes incorporates a]”...strengths-based approach [by] really looking at the

person’s own capacity and their own aspirations and seeing how you can… incorporate that into their recovery process” (Participant 9).

On Page 10, you wrote:

Activities like that, is where you can speak with a client in a casual…setting rather than a structured formalised assessment or an interview room, i think breaks down power structures…” (Participant 11).

Change to:

Activities like that, is where you can speak with a client in a casual…setting rather than a structured formalised assessment or an interview room, I think breaks down power structures…” (Participant 11).

On Page 12, you wrote:

This resonates with Sapouna & Pamer’s (2016) research who identified that many mental health consumers involved in digital storytelling projects, spoke of a broadening of their self-concept beyond a narrowly defined ‘patient’, or someone with ‘ill health’. Such accounts provide a positive counter narrative disrupting the dominant stigmatising narratives often present in stigmatising popular media representations Ross, Morgan, Form & Revley, 2019), particularly for consumers, families and carers involved in forensic mental health.

Change to:

This resonates with Sapouna and Pamer’s (2016) research who identified that many mental health consumers involved in digital storytelling projects, spoke of a broadening of their self-concept beyond a narrowly defined ‘patient’, or someone with ‘ill health’. Such accounts provide a positive counter narrative disrupting the dominant stigmatising narratives often present in stigmatising popular media representations (Ross et al., 2019), particularly for consumers, families and carers involved in forensic mental health.

On Page 13, you wrote:

owing to the COVID-19 pademic.

Change to:

owing to the COVID-19 pandemic.

OTHER ISSUES:

·        I would like you to provide more demographic information regarding your participants (i.e., gender, age, race, socioeconomic status, length of time as a social worker, etc.)

THERE ARE SEVERAL APA ISSUES WITHIN THE PAPER AND ON THE REFERENCE PAGE. THE AUTHORS MUST CORRECT THESE ERRORS.

According to the 7th edition of APA:

1.       You should alphabetize in-text citations – You made this error several times in the manuscript

2.       You should alphabetize citations [See Davey & Dempsey, 2012; Davidson et al., 2018. These citations are not in the proper order.]

3.       You should capitalize and italicize journal titles

4.       You should italicize the volume number – You made this error several times on the Reference page

5.       You should not italicize the issue number

6.       You should keep the volume number AND issue number together 

You wrote:

Brown C. (2021) Critical clinical social work and the neoliberal constraints on social justice in mental health in Research on Social Work Prctice, 31 (6) 644 – 652

Change to:

Brown C. (2021) Critical clinical social work and the neoliberal constraints on social justice in mental health. Research on Social Work Practice, 31(6) 644-652.

7.       You should italicize book titles

8.       You should capitalize journal titles

9.       You should provide beginning and ending page numbers

10.     When citing 3 or more authors, you should use the surname of the first author followed by et al and the year [This is the case for in-text citations and citations within a sentence] – You made this error several times in the manuscript

11.     You should not capitalize every word in your title. [Only capitalize the proper nouns]

12.     Add a period behind the initial of an author’s name.

You wrote this:

Davidson T, Moreland A, Bunnell B.E, Winkelmann J, Hamblen J.L. & Ruggiero K.J (2018) Chapter 7 Reducing Stigma in Mental Health Through Storytelling in Deconstructing Stigma in Mental Health, Canfield B.A. & Cunningham H.A. (Eds), IGI Global, USA

Change to:

Davidson T., Moreland, A., Bunnell, B. E., Winkelmann, J., Hamblen, J. L., & Ruggiero, K. J. (2018). B. A. Canfield & H. A. Cunnington (Eds.) in Reducing Stigma in Mental Health Through Storytelling in Deconstructing Stigma in Mental Health (pp. XX – XX). IGI Global, USA.

·        Notice how I added a space, a period, and a comma after the initial of each author’s name. Do the same for the other citations.

·        Notice how I added & before the surname and initial of the last author. Do the same for the other citations.

·        Add the beginning and ending pages for this chapter (pp. XX – XX).

13.     Add a period at the end of your citation (specifically after the ending page number). [The only exception is when you provide the full URL for websites]

This article reports on findings of a study that explores the utility of digital storytelling as a narrative practice and learning tool for social work in an Australian secure forensic mental health hospital. The short digital stories, or Digital Bytes project, centred on capturing the lived experience, hopes and perspectives of the hospital’s service users by giving voice to their experiences through digital technology. The project was collaboratively designed and co-delivered with social work students, hospital staff, and service users. It aimed to not only destigmatise people with lived experiences of mental distress and criminal justice system involvement, but to also give staff and students further insights into understanding who they are working with.

The authors provided a work that was informative, well written, and provides a solid foundation for future research. I enjoyed reading this manuscript, and recommend the authors thoroughly read the manuscript and correct any spelling and/or grammatical errors. It is in the spirit of improving the manuscript that I offer the following comments and/or questions:

Change your title to:

The Digital Bytes Project: Digital Storytelling as a Tool for Challenging Stigma and Making Connections in a Forensic Mental Health Setting

On Page 1, you wrote: Digital Bytes project 

Change all to: Digital Bytes Project [Sometimes you spell “project” and other times you spell “Project” – Make it consistent throughout]

On Page 1:

Change “centred” to “centered.”

On Page 2, you wrote:

While the benefits and risks of using DST to support mental health recovery have been relatively well explored (see De Vecchi et al. 2016, Botfield et al. 2017, and de Jager et al. 2017), its utility within a forensic mental health context appears unexamined. This is significant, given the so-called ‘dual stigma’ experienced by people involved in both forensic and mental health systems (see Marklund, Wahloos, Loori & Gabrielsson, 2020; West, Yanos & Mulay, 2014; O'Donahoo & Simmonds, 2016).

Change to:

While the benefits and risks of using DST to support mental health recovery have been relatively well explored (see Botfield et al., 2017; de Jager et al., 2017; De Vecchi et al., 2016), its utility within a forensic mental health context appears unexamined. This is significant, given the so-called ‘dual stigma’ experienced by people involved in both forensic and mental health systems (Marklund et al., 2020; O'Donahoo & Simmonds, 2016; West et al., 2014).

·        Per the 7th Edition of APA, in-text citations should be in alphabetical order.

·        Per the 7th Edition of APA, when citing 3 or more authors, you should use the surname of the first author, et al., and then the year.

·        Make sure that all citations that are in the manuscript are also on the Reference page [See Botfield et al., 2017; de Jager et al., 2017; De Vecchi et al., 2016; Marklund et al., 2020; O'Donahoo & Simmonds, 2016; West et al., 2014]. Several citations were in the manuscript but not on the Reference page.

NOTE: You made these errors several times in the manuscript. Please correct these errors.

On Page 3, you wrote:

A scoping review was conducted of peer reviewed literature published between 2011 – 2021, to examine how digital storytelling (DST) is being used to address experiences of stigma and discrimination for mental health service users, particularly forensic mental health service users (Lambert, Egan, Turner & Martin, 2023 - under review).

·        Why did you start in the year 2011? Is this the year that digital storytelling (DST) was first used among mental health service users, specifically forensic mental health service users?

On Page 4, you wrote:

Braun and Clarke's (2006) six-step framework for conducting a thematic analysis was used to analyse the interview transcripts. Our researchers analysed and coded each transcript independently, then compared codes once all interview transcripts had been coded to identify key themes and subthemes.

·        This citation (Braun & Clark, 2006) should be added to the Reference page.

·        Make sure that all citations that are in the manuscript are also on the Reference page.

On Page 4, you wrote:

Our researchers analysed and coded each transcript independently, then compared codes once all interview transcripts had been coded to identify key themes and subthemes.

·        Did you establish reliability between the coders? If so, what was the reliability percentage? If not, didn’t you establish reliability between the coders?

On Page 4, you wrote:

This stigma involves a deep discrediting and devaluing of persons living with mental ill-health by others” (Groot, 2021, p.7).

Change to:

This stigma involves a deep discrediting and devaluing of persons living with mental ill-health by others” (Groot, 2021, p. 7).

·        I added a space between p. and 7.

On Page 8, you wrote:

“There was one [digital byte] that I watched…that was quite worthwhile because…this patient had a particularly… for want of a better word ‘notorious’ history and background… Watching the video… was really worthwhile in terms of learning what type of things i’d be able to engage with them on if I ever had to. It… also broadened the scope of actual recovery-based activities that could be done together. I had no idea that this person enjoyed… cooking and sculpture… I feel like that sort of knowledge, I wouldn’t have known… unless I watched [their digital byte]. Activities like that… where you can speak with a client in a casual…setting, rather than a structured formalised assessment or an interview room – i think breaks down power structures…” (Participant 11).

Change to:

“There was one [digital byte] that I watched…that was quite worthwhile because…this patient had a particularly… for want of a better word ‘notorious’ history and background… Watching the video… was really worthwhile in terms of learning what type of things I’d be able to engage with them on if I ever had to. It… also broadened the scope of actual recovery-based activities that could be done together. I had no idea that this person enjoyed… cooking and sculpture… I feel like that sort of knowledge, I wouldn’t have known… unless I watched [their digital byte]. Activities like that… where you can speak with a client in a casual…setting, rather than a structured formalised assessment or an interview room – I think breaks down power structures…” (Participant 11).

On Page 9, you wrote:

[Digital bytes incorporates a]”...strengths-based approach [by] really looking at the

person own capacity and their own aspirations and seeing how you can… incorporate that into their recovery process” (Participant 9).

Change to:

[Digital bytes incorporates a]”...strengths-based approach [by] really looking at the

person’s own capacity and their own aspirations and seeing how you can… incorporate that into their recovery process” (Participant 9).

On Page 10, you wrote:

Activities like that, is where you can speak with a client in a casual…setting rather than a structured formalised assessment or an interview room, i think breaks down power structures…” (Participant 11).

Change to:

Activities like that, is where you can speak with a client in a casual…setting rather than a structured formalised assessment or an interview room, I think breaks down power structures…” (Participant 11).

On Page 12, you wrote:

This resonates with Sapouna & Pamer’s (2016) research who identified that many mental health consumers involved in digital storytelling projects, spoke of a broadening of their self-concept beyond a narrowly defined ‘patient’, or someone with ‘ill health’. Such accounts provide a positive counter narrative disrupting the dominant stigmatising narratives often present in stigmatising popular media representations Ross, Morgan, Form & Revley, 2019), particularly for consumers, families and carers involved in forensic mental health.

Change to:

This resonates with Sapouna and Pamer’s (2016) research who identified that many mental health consumers involved in digital storytelling projects, spoke of a broadening of their self-concept beyond a narrowly defined ‘patient’, or someone with ‘ill health’. Such accounts provide a positive counter narrative disrupting the dominant stigmatising narratives often present in stigmatising popular media representations (Ross et al., 2019), particularly for consumers, families and carers involved in forensic mental health.

On Page 13, you wrote:

owing to the COVID-19 pademic.

Change to:

owing to the COVID-19 pandemic.

OTHER ISSUES:

·        I would like you to provide more demographic information regarding your participants (i.e., gender, age, race, socioeconomic status, length of time as a social worker, etc.)

THERE ARE SEVERAL APA ISSUES WITHIN THE PAPER AND ON THE REFERENCE PAGE. THE AUTHORS MUST CORRECT THESE ERRORS.

According to the 7th edition of APA:

1.       You should alphabetize in-text citations – You made this error several times in the manuscript

2.       You should alphabetize citations [See Davey & Dempsey, 2012; Davidson et al., 2018. These citations are not in the proper order.]

3.       You should capitalize and italicize journal titles

4.       You should italicize the volume number – You made this error several times on the Reference page

5.       You should not italicize the issue number

6.       You should keep the volume number AND issue number together 

You wrote:

Brown C. (2021) Critical clinical social work and the neoliberal constraints on social justice in mental health in Research on Social Work Prctice, 31 (6) 644 – 652

Change to:

Brown C. (2021) Critical clinical social work and the neoliberal constraints on social justice in mental health. Research on Social Work Practice, 31(6) 644-652.

7.       You should italicize book titles

8.       You should capitalize journal titles

9.       You should provide beginning and ending page numbers

10.     When citing 3 or more authors, you should use the surname of the first author followed by et al and the year [This is the case for in-text citations and citations within a sentence] – You made this error several times in the manuscript

11.     You should not capitalize every word in your title. [Only capitalize the proper nouns]

12.     Add a period behind the initial of an author’s name.

You wrote this:

Davidson T, Moreland A, Bunnell B.E, Winkelmann J, Hamblen J.L. & Ruggiero K.J (2018) Chapter 7 Reducing Stigma in Mental Health Through Storytelling in Deconstructing Stigma in Mental Health, Canfield B.A. & Cunningham H.A. (Eds), IGI Global, USA

Change to:

Davidson T., Moreland, A., Bunnell, B. E., Winkelmann, J., Hamblen, J. L., & Ruggiero, K. J. (2018). B. A. Canfield & H. A. Cunnington (Eds.) in Reducing Stigma in Mental Health Through Storytelling in Deconstructing Stigma in Mental Health (pp. XX – XX). IGI Global, USA.

·        Notice how I added a space, a period, and a comma after the initial of each author’s name. Do the same for the other citations.

·        Notice how I added & before the surname and initial of the last author. Do the same for the other citations.

·        Add the beginning and ending pages for this chapter (pp. XX – XX).

13.     Add a period at the end of your citation (specifically after the ending page number). [The only exception is when you provide the full URL for websites]

Round 2

Reviewer 3 Report

First, I commend the authors for the clarity of their revisions. Using red text made it easy to determine the revisions made. With that said, I strongly recommend that the authors do a thorough read of the manuscript and correct any issues in spelling or grammar. For example, on Page 10, the authors wrote,

However, this study provides useful insights into the utility of the Digital Bytes Project for social work practice in forensic mental health and highlights the natural alignment between the social justice values that underpin both social work and lived experience and the narrative foundations of digital storytelling. This research also does not explore the utility of the Digital  Bytes ProjectDigital Bytes Projects of consumers, or their family or carers, and this is essential for measuring or understanding utility in a more meaningful and holistic way. 

In this passage, the authors use the word "Digital Bytes Project" and "Digital Bytes Projects" twice. I assume that you only intend to use one, correct? 

In addition, there are still inconsistencies with the citations on your Reference page. 

First, I commend the authors for the clarity of their revisions. Using red text made it easy to determine the revisions made. With that said, I strongly recommend that the authors do a thorough read of the manuscript and correct any issues in spelling or grammar. For example, on Page 10, the authors wrote,

However, this study provides useful insights into the utility of the Digital Bytes Project for social work practice in forensic mental health and highlights the natural alignment between the social justice values that underpin both social work and lived experience and the narrative foundations of digital storytelling. This research also does not explore the utility of the Digital  Bytes Project Digital Bytes Projects of consumers, or their family or carers, and this is essential for measuring or understanding utility in a more meaningful and holistic way. 

In this passage, the authors use the word "Digital Bytes Project" and "Digital Bytes Projects" twice. I assume that you only intend to use one, correct? 

In addition, there are still inconsistencies with the citations on your Reference page. 
